# General-Purpose In-Context Learning by Meta-Learning Transformers

## Abstract

Modern machine learning requires system designers to specify aspects of the learning pipeline, such as losses, architectures, and optimizers. Meta-learning, or learning-to-learn, instead aims to *learn* those aspects, and promises to unlock greater capabilities with less manual effort. One particularly ambitious goal of meta-learning is to train general-purpose in-context learning algorithms from scratch, using only black-box models with *minimal inductive bias*. Such a model takes in training data, and produces test-set predictions across a wide range of problems, without any explicit definition of an inference model, training loss, or optimization algorithm. In this paper we show that Transformers and other black-box models can be meta-trained to act as general-purpose in-context learners. We characterize phase transitions between algorithms that generalize, algorithms that memorize, and algorithms that fail to meta-train at all, induced by changes in model size, number of tasks, and meta-optimization. We further show that the capabilities of meta-trained algorithms are bottlenecked by the accessible state size (memory) determining the next prediction, unlike standard models which are thought to be bottlenecked by parameter count. Finally, we propose practical interventions such as biasing the training distribution that improve the meta-training and meta-generalization of general-purpose in-context learning algorithms.

## 1 Introduction

Meta-learning is the process of automatically *discovering new learning algorithms* instead of designing them manually (Schmidhuber, 1987). An important quality of human-engineered learning algorithms, such as backpropagation and gradient descent, is their applicability to a wide range of tasks or environments. For learning-to-learn to exceed those capabilities, the meta-learned learning algorithms must be similarly *general-purpose*. Recently, there has been significant progress toward this goal (Kirsch et al., 2019; Oh et al., 2020). The improved generality of the discovered learning algorithms has been achieved by introducing inductive bias, such as by bottlenecking the architecture or by hiding information, which encourage learning over memorization. Methods include restricting learning rules to use gradients (Metz et al., 2019; Kirsch et al., 2019; Oh et al., 2020), symbolic graphs (Real et al., 2020; Co-Reyes et al., 2021), or parameter sharing (Kirsch & Schmidhuber, 2020; Kirsch et al., 2021).

While enabling generalization, these inductive biases come at the cost of increasing the effort to design these systems and potentially restrict the space of discoverable learning algorithms. Instead, we seek to explore general-purpose meta-learning systems with *minimal inductive bias*. Good candidates for this are black-box sequence-models as meta-learners such as LSTMs (Hochreiter et al., 2001; Wang et al., 2016; Duan et al., 2016) or Transformers (Vaswani et al., 2017). These memory-based or in-context learners take in training data and produce test-set predictions without any explicit definition of an inference model, training loss, or optimization algorithm. With recent advances of in-context learning in large language models (Brown et al., 2020), neural networks can already learn many concepts from demonstrations. What are the necessary conditions such that those models can learn from a wide range of demonstrations? To what extent can we elicit in-context learning that generalizes to a wider range of problems, in a similar way how learning via backpropagation and gradient descent can generalize?

In this work, we investigate how such in-context meta-learners can be trained to (meta-)generalize and learn on significantly different datasets than used during meta-training. For this we propose a Transformer-based *General-Purpose In-Context Learner* (GPICL) which is described with an associated meta-training task distribution in Section 3. In Section 4.1 we characterize phase transitions—induced by scaling the number of tasks or the model size used for meta-training—between memorization, task identification, and general learning-to-learn. We further show in Section 4.2 that the capabilities of meta-trained algorithms are bottlenecked by their accessible state (memory) size determining the next prediction (such as the hidden state size in a recurrent network), unlike standard models which are thought to be bottlenecked by parameter count. Finally, in Section 4.3, we propose practical interventions that improve the meta-training of general purpose learning algorithms.

## 2    Background

**What is a (supervised) learning algorithm?** In this paper, we focus on the setting of meta-learning supervised in-context learning algorithms. Consider a mapping

$$\left( \{x_i, y_i\}_{i=1}^{N_D}, x' \right) \mapsto y' \tag{1}$$

from the training (support) set $D = \{x_i, y_i\}_{i=1}^{N_D}$ and a query input $x'$ to the query's prediction $y'$ where $x_i, x' \in \mathbb{R}^{N_x}$, $y_i, y' \in \mathbb{R}^{N_y}$ and $N_D, N_x, N_y \in \mathbb{N}^+$. The subset of these functions that qualify as learning algorithms are those that improve their predictions $y'$ given an increasingly larger training set $D$. Meta-learning then corresponds to finding these functions via meta-optimization. As in other black-box meta-learning models, we use a neural network to represent such functions. Such in-context learning is different from gradient-based meta-learning (such as MAML (Finn et al., 2017)) in that no explicit gradients are computed at meta-test time. All required mechanisms for learning are implicitly encoded in the black-box neural network.

**What is a general-purpose learning algorithm?** A learning algorithm can be considered general-purpose if it learns on a wide range of possible tasks $D$ and their respective related queries $x', y'$. In this paper, we are interested in strong generalization across entirely different datasets such as MNIST, Fashion MNIST, and CIFAR10. Human-engineered learning algorithms such as gradient-descent on a suitable loss function can be considered general-purpose learning algorithms that can be applied to any of these datasets (where the gradient is obtained via backpropagation or other means). Meta-learners often don't generalize that well at meta-test time when we have an entirely new dataset that we want to learn on. We set out to investigate under which conditions in-context learning generalizes well. In comparison to in-context learning, gradient-based methods like MAML hard-code the human-engineered learning algorithm of gradient descent and inherit its generalization properties.

## 3    General-Purpose In-Context Learning

Due to the small number of inductive biases in black-box models, we can only expect (meta-)generalization when meta-training with an appropriately broad data distribution. Thus, changes in the data distribution affect whether and how a model meta-learns and meta-generalizes. We classify algorithms along two different dimensions: To what extent it learns (improving predictions given increasingly larger training sets provided at inference time), and to what extent it generalizes (performs well on instances, tasks, or datasets not seen before). Algorithms can then be categorized as in Table 1. In task memorization, the model immediately performs well on seen tasks but does not gen-

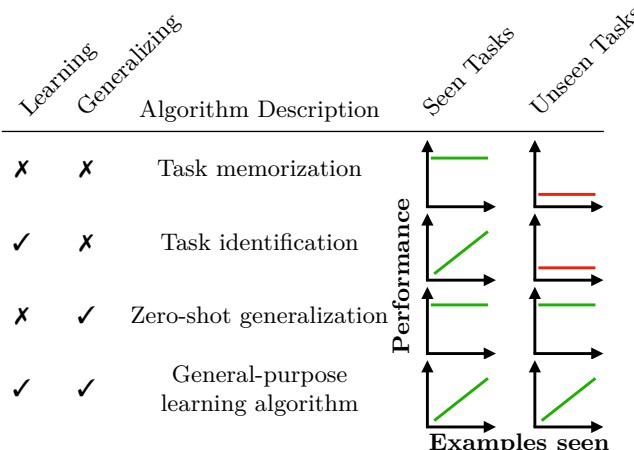

Table 1: An algorithm encoded in a neural network can be classified along two different dimensions: To what extent it *learns* and to what extent it *generalizes*.

eralize. In task identification, the model identifies the task and gets better on it at inference time as it sees more examples but can only do so on tasks very similar to what it was trained on. In zero-shot generalization, the model immediately generalizes to unseen tasks, without observing examples. Finally, a general-purpose learning algorithm improves as it observes more examples both on seen and significantly different unseen tasks. We demonstrate that sharp phase transitions occur between these learning modalities, and empirically investigate these.

### 3.1 Generating Tasks for Learning-to-Learn

Neural networks are known to require datasets of significant size to effectively generalize. While in standard supervised learning large quantities of data are common, meta-learning algorithms may require a similar number of distinct *tasks* in order to learn and generalize. Unfortunately, the number of commonly available tasks is orders of magnitudes smaller compared to the datapoints in each task.

Previous work has side-stepped this issue by building-in architectural or algorithmic structure into the learning algorithm, in effect drastically reducing the number of tasks required. For example, in Kirsch & Schmidhuber (2020); Kirsch et al. (2021), the authors included symmetries into the black-box model in the form of input and output permutation invariances. An alternative to this is the generation of new tasks (Schmidhuber, 2013; Clune, 2019; Such et al., 2020; Parker-Holder et al., 2022). Unfortunately, it is not easy to generate a wide range of tasks that are both diverse and contain structure as it can be found in the real world.

In this work, we take an intermediate step by augmenting existing datasets, in effect increasing the breadth of the task distribution based on existing task regularities. We generate a large number of tasks by taking existing supervised learning datasets, randomly projecting their inputs and permuting their classification labels. While the random projection removes spatial structure from the inputs, this structure is not believed to be central to the task (for instance, the performance of SGD-trained fully connected networks is invariant to projection by a random orthogonal matrix (Wadia et al., 2021)). Task augmentation allows us to investigate fundamental questions about learning-to-learn in the regime of many

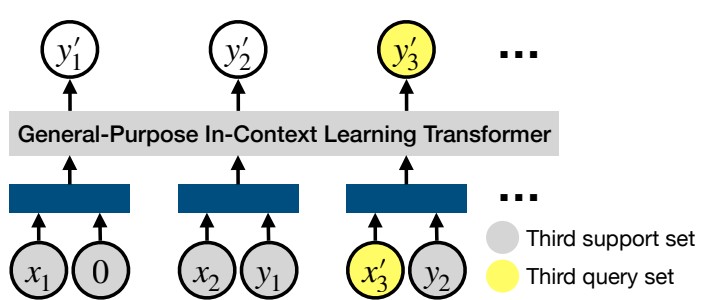

Figure 1: Our *General-Purpose In-Context Learner* (GPICL) is based on the vanilla Transformer which is trained to make predictions for queries $x'$ given any prefix of a dataset $D := \{x_i, y_i\}_{i=1}^{N_D}$ as in Equation 2.

tasks without relying on huge amounts of existing tasks or elaborate schemes to generate those.

A task or dataset $D$ is then defined by its corresponding base dataset $\bar{D} = \{\bar{x}_i, \bar{y}_i\}$, (linear) projection $A \in \mathbb{R}^{N_x \times N_x}$, with $A_{ij} \sim \mathcal{N}\left(0, \frac{1}{N_x}\right)$, and output permutation $\rho$, $D = \{A\bar{x}_i, \rho(\bar{y}_i)\}$. Unless noted otherwise, the distribution over output permutations $p(\rho)$ is uniform.

### 3.2 Meta-learning and meta-testing

**Meta-learning** Given those generated tasks, we then meta-train jointly on a mini-batch sampled from the whole distribution. First, we sample datasets $D$ from the augmented task distribution $p(D)$ and then take a random batch $D_{1:N_D}$ from the training set. Second, we minimize $J(\theta)$, the sum of losses on the query prediction after observing any prefix $D_{1:j-1}$

$$J(\theta) = \mathbb{E}_{D \sim p(D)} \left[ \sum_{j=1}^{N_D} l(f_\theta(D_{1:j-1}, x_j), y_j) \right], \tag{2}$$

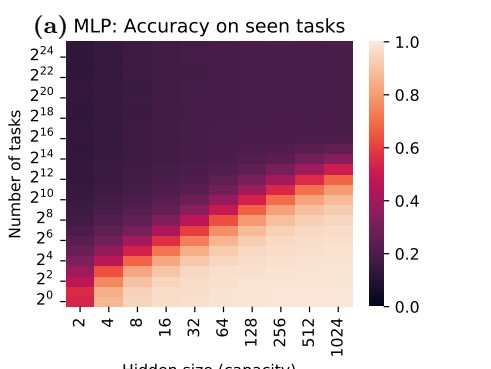 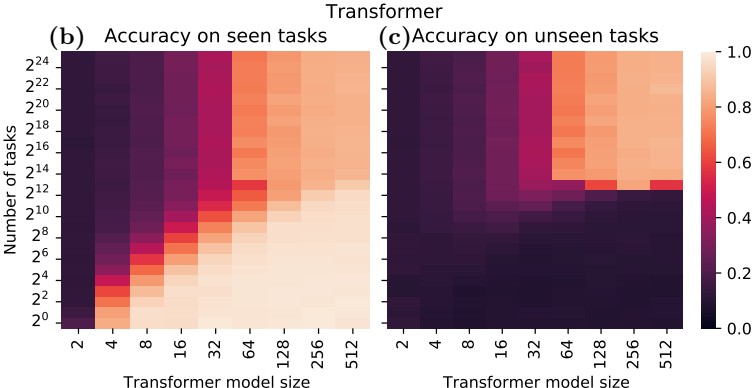

Figure 2: **GPICL is able to generalize to unseen tasks.** Each cell is a separate meta-training run. **(a)** An MLP classifier trained in a multi-task fashion across various numbers of tasks (generated based on MNIST) and network sizes is able to fit linearly more tasks, the larger its capacity. **(b)** A sequence model (here the GPICL Transformer) that observes a dataset $D$ of inputs and labels transitions into generalizing to an seemingly unbounded number of tasks with an increase in model size. This is achieved by switching from a memorization solution to a learning solution that **(c)** generalizes to unseen tasks. This generalization does not occur with the MLP.

---

**Algorithm 1** Meta-Training for General-Purpose In-Context Learning (GPICL) via Augmentation

---

**Require:** Dataset $\bar{D} = \{\bar{x}_i, \bar{y}_i\}$, Number of tasks $K \in \mathbb{N}^+$
 # Define $p(D)$ by augmenting $\bar{D}$, here by:
 $\{A_{ij}^{(k)}\}_{k=1}^{K} \sim \mathcal{N}(0, \frac{1}{N_x})$              ▷ Sample input projections
 $\{\rho^{(k)}\}_{k=1}^{K} \sim p(\rho)$               ▷ Sample output permutations
 $D^{(k)} = \{A^{(k)}\bar{x}_i, \rho^{(k)}(\bar{y}_i)\}$
 $p(D) := \text{Uniform}[\{D^{(k)}\}_{k=1}^{K}]$

 # Meta-Training on $p(D)$
 **while** not converged **do**
   $\theta \leftarrow \theta - \alpha \nabla_\theta J(\theta)$                   ▷ Equation 2

---

where in the classification setting, $l$ is the cross entropy loss between the label $y_j$ and prediction $y' = f_\theta(D_{1:j-1}, x_j)$, $f_\theta$ is a neural network mapping to predictions $y'$ as in Equation 1. During meta-training, we take gradient steps in $J(\theta)$ by backpropagation and Adam (Kingma & Ba, 2014). To investigate the effect of the data distribution, we train on various numbers of tasks (Algorithm 1). Finally, we need to choose a black-box model for the function $f_\theta$. We use a vanilla Transformer (Vaswani et al., 2017) with learned positional embeddings, visualized in Figure 1. We call it the *General-Purpose In-Context Learner* (GPICL). Each token corresponds to the concatenation of a transformed input $x_i$ and one-hot encoded label $y_{i-1}$. The model predicts the corresponding logits $y' = y_i$ for the current input $x' = x_i$. When querying for the first $x_1$, no label for the previous input is available, so we feed a zero vector.

**Meta-testing** At meta-test time, no gradient-based learning is used. Instead, we simply obtain a prediction $y'$ by evaluating the neural network $f_\theta$ on a dataset $D$ and query point $x'$. The dataset $D$ is either derived from the same base dataset (eg MNIST after meta-training on MNIST) or it is derived from a different dataset (eg Fashion MNIST or CIFAR10). In both cases a seen or unseen random projection is used. Datapoints are taken only from the respective test split of the base dataset.

## 4   Experiments on the Emergence of General Learning-To-Learn

**Multi-task training with standard classifiers** Given a task distribution of many different classification tasks, we first ask under what conditions we expect "learning-to-learn" to emerge. We train a single model

across many tasks where each task corresponds to a random transformation of the MNIST dataset, but where the MLP only receives a single datapoint instead of a whole sequence as input. This corresponds to $N_D = 1$ in Equation 2. We would expect such a non-sequential classifier to be able to correctly predict for more tasks as its number of parameters increases. When plotting the network capacity against the number of tasks, we indeed observe a linear boundary where an increasing number of tasks can be fit the larger the network (Figure 2a). This is consistent with results in Collins et al. (2016), which found that a constant number of bits about the data distribution can be stored per model parameter, across a variety of model architectures and scales.

**Learning-to-learn with large sequential models and data** In contrast to the MLP classifier, a sequence model that observes multiple observations and their labels from the same task, could exceed that linear performance improvement by learning at inference time. Indeed, we observe that when switching to a Transformer that can observe a sequence of datapoints before making a prediction about the query, more tasks can be simultaneously fit (Figure 2b). At a certain model size and number of tasks, the model undergoes a phase transition, allowing to generalize to a seemingly unbounded number of tasks. We hypothesize that this is due to switching the prediction strategy from memorization to learning-to-learn. Further, when (meta-)testing the same trained models from the previous experiment on an unseen task (new random transformation of MNIST), they generalize only in the regime of large numbers of tasks and model size (Figure 2c). As an in-context learner, meta-testing does not involve any gradient updates but only running the model in forward mode.

**Insight 1: It is possible to learn-to-learn with black-box models** Effective learning algorithms can be realized in-context using black-box models with few inductive biases, given sufficient meta-training task diversity and large enough model sizes. To transition to the learning-to-learn regime, we needed at least $2^{13} = 8192$ tasks.

In the following, we study learning-to-learn from the perspective of the *data distribution*, the *architecture*, and the *optimization dynamics*. For the data distribution, we look at how the data diversity affects the emergence and phase transitions of learning-to-learn, generalization, and memorization. For architecture, we analyze the role of the model and state size in various architectures. Finally, we observe challenges in meta-optimization and demonstrate how memorization followed by generalization is an important mechanism that can be facilitated by biasing the data distribution.

### 4.1 Large Data: Generalization and Phase Transitions

**Simple data augmentations lead to the emergence of learning-to-learn** To verify whether the observed generalizing solutions actually implement learning algorithms (as opposed to e.g. zero-shot generalization), we analyze the meta-test time behavior. We plot the accuracy for a given query point for varying numbers of examples in Figure 3. As is typical for learning algorithms, the performance improves when given more examples (inputs and labels).

**Generalization** Naturally, the question arises as to what extent these learning algorithms are general. While we have seen generalization to unseen tasks consisting

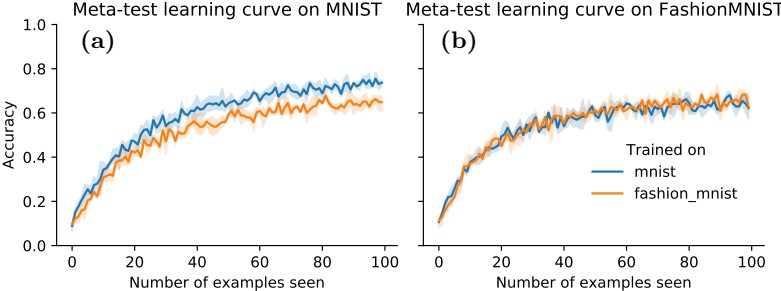

Figure 3: **GPICL learns from examples at test time, and generalizes to unseen tasks and datasets.** We meta-trained the Transformer on a set of tasks defined by random transformations of either MNIST (blue) or FashionMNIST (orange). We then meta-test on unseen tasks, and seen (ab) or unseen (ba) datasets. The plot shows the accuracy averaged across multiple runs at each inner step, with shading indicating 95% confidence intervals. The increase in performance at each step suggests we have learned a learning algorithm.

Table 2: Meta-test generalization to various datasets after meta-training on augmented MNIST and seeing 99 examples, predicting the 100th. We report the mean across 3 meta-training seeds, 16 sequences from each task, 16 tasks sampled from each base dataset. GPICL is competitive to other approaches that require more inductive bias.

| Method / Dataset | Inductive bias | MNIST | Fashion MNIST | KMNIST | Random | CIFAR10 | SVHN |
|---|---|---|---|---|---|---|---|
| SGD | Backprop, SGD | 70.31% | 50.78% | 37.89% | 100.00% | 14.84% | 10.16% |
| MAML | Backprop, SGD | 53.71% | 48.44% | 36.33% | 99.80% | 17.38% | 11.33% |
| VSML | In-context, param sharing | 79.04% | 68.49% | 54.69% | 100.00% | 24.09% | 17.45% |
| LSTM | In-context, black-box | 25.39% | 28.12% | 18.10% | 58.72% | 12.11% | 11.07% |
| GPICL Transformer (ours) | In-context, black-box | 73.70% | 62.24% | 53.39% | 100.00% | 19.40% | 14.58% |

of novel projections of the same dataset, do the learned algorithms also generalize to unseen datasets? In Figure 3 we observe strong out-of-distribution performance on Fashion MNIST after having trained on MNIST (b, blue), and there is no generalization gap compared to directly training on Fashion MNIST (b, orange). Similarly, when meta training on Fashion MNIST and meta testing on MNIST (a, orange) we observe that the learning algorithm generalizes, albeit with a larger generalization gap.

**Comparison to other methods** Other datasets and baselines are shown in Table 2. We aim to validate whether methods with less inductive bias (such as our GPICL), can compete with methods that include more biases suitable to learning-to-learn. This includes stochastic gradient descent (SGD), updating the parameters online after observing each datapoint. MAML (Finn et al., 2017) proceeds like SGD, but uses a meta-learned neural network initialization. Both methods that rely on backpropagation and gradient descent learn more slowly than our Transformer. In the case of MAML, this may be due to the main mechanism being feature reuse (Raghu et al., 2020) which is less useful when training across our wider task distribution. For in-context learners (methods that do not hard-code gradient descent at meta-test time), we test VSML (Kirsch & Schmidhuber, 2020) that discovered learning algorithms significantly generalizing between tasks. Our GPICL comes surprisingly close to VSML without requiring the associated inductive bias. GPICL generalizes to many datasets, even those that consist of random input-label pairs.

We also observe that learning CIFAR10 and SVHN from only 99 examples with a general-purpose learning algorithm is difficult, which we address in Section 4.4. Training and testing with longer context lengths improves the final predictions (Appendix A.2). Using LSTM-based in-context learners performs worse, which we further discuss in Section 4.2 among other alternative architectures.

**Insight 2: Simple data augmentations are effective for learning-to-learn** The generality of the discovered learning algorithm can be controlled via the data distribution. Even when large task distributions are not (yet) naturally available, simple augmentations are effective.

**Transitioning from memorization to task identification to general learning-to-learn** When do the learned models correspond to memorizing, learning, and generalizing solutions? In Figure 4, we meta-train across varying numbers of tasks, with each point on the x-axis corresponding to multiple separate meta-training runs. We plot the accuracy difference between the last and first prediction (how much is learned at meta-test time) for a seen task, an unseen task, and an unseen task

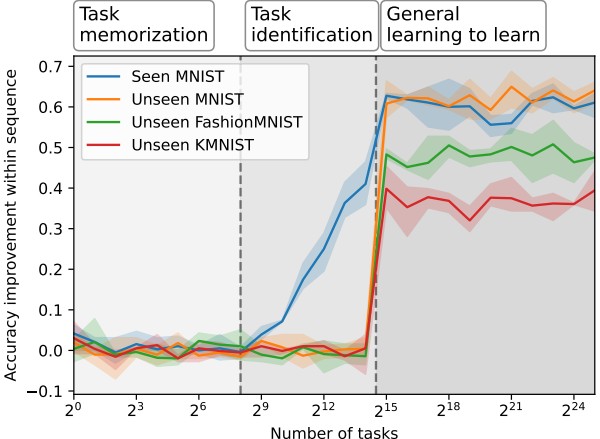

Figure 4: **Transformers exhibit three different phases in terms of meta-learned behavior.** (1) When training on a small number of tasks, tasks are memorized. (2) Tasks from the training distribution are identified, which is evident as a within-sequence increase of performance. (3) When training across many tasks, we discover a learning algorithm that generalizes to unseen tasks and unseen datasets.

with a different base dataset. We observe three phases: In the first phase, the model memorizes all tasks, resulting in no within-sequence performance improvement. In the second phase, it memorizes and learns to identify tasks, resulting in a within-sequence improvement confined to seen task instances. In the final and third phase, we observe a more general learning-to-learn, a performance improvement for unseen tasks, even different base datasets (here FashionMNIST). This phenomenon applies to various other meta-training and meta-testing datasets. The corresponding experiments can be found in Appendix A.5.

**The phase transition to general learning-to-learn** In Figure 4 we observe a fairly discrete transition from task identification to generalizing learning-to-learn (the second dashed line) as a function of the number of tasks. Previously, Figure 2 (c) showed a similarly discrete phase transition from no learning to learning on unseen tasks. What happens during this transition and when do the found solutions correspond to memorizing (task memorization or seen task identification) vs generalizing solutions? To analyze the transition from task identification to general learning to learn, we perform multiple training runs with varying seeds and numbers of tasks on MNIST. This is shown in Figure 5, reporting the final training loss. We find that the distribution is bi-modal. Solutions at the end of training are memorizing or generalizing. Memorization cluster: The larger the number of tasks, the more difficult it is to memorize all of them with a fixed model capacity (or learn to identify each task). Generalization cluster: At a certain number of tasks (here 6000), a transition point is reached where optimization sometimes discovers a lower training loss that corresponds to a generalizing learning to learn solution. For larger numbers of tasks the solutions always settle in the generalizing cluster.

**Insight 3: The meta-learned behavior has phase transitions** When increasing the number of tasks, the meta-learned behavior transitions from task memorization, to task identification, to general learning-to-learn. The last transition is discrete, with two unique clusters.

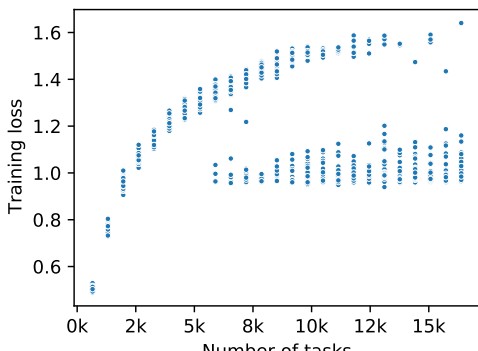

Figure 5: **Solutions found by GPICL after meta-training are bi-modal, with a memorization and generalization mode.** Each point represents the training loss at the end of meta-training for runs with different seeds and for various numbers of tasks that include the transition boundary previously observed. Almost all solutions are either in a memorization cluster or in a generalization cluster.

## 4.2 Architecture: Large Memory (State) is Crucial for Learning

In the previous experiments we observed that given sufficient task diversity and model size, Transformers can learn general-purpose learning algorithms. This raises the question how essential the Transformer architecture is and whether other black-box models could be used. We hypothesize that for learning-to-learn the size of the memory at meta-test time (or state more generally) is particularly important in order to be able to store learning progress. Through self-attention, Transformers have a particularly large state. We test this by training several architectures with various state sizes in our meta-learning setting. In Figure 6a, we observe that when we vary the hyper-parameters which most influence the state size, we observe that for a specific state size we obtain similar performance of the discovered learning algorithm across architectures. In contrast, these architectures have markedly different numbers of parameters (Figure 6b).

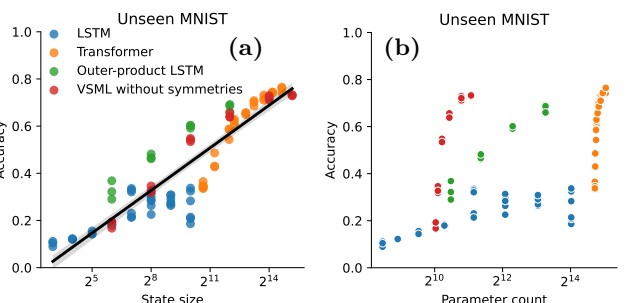

Figure 6: **The state size (accessible memory) of an architecture most strongly predicts its performance as a general-purpose learning algorithm.** **(a)** A large state is crucial for learning-to-learn to emerge. **(b)** The parameter count correlates less well with learning capabilities.

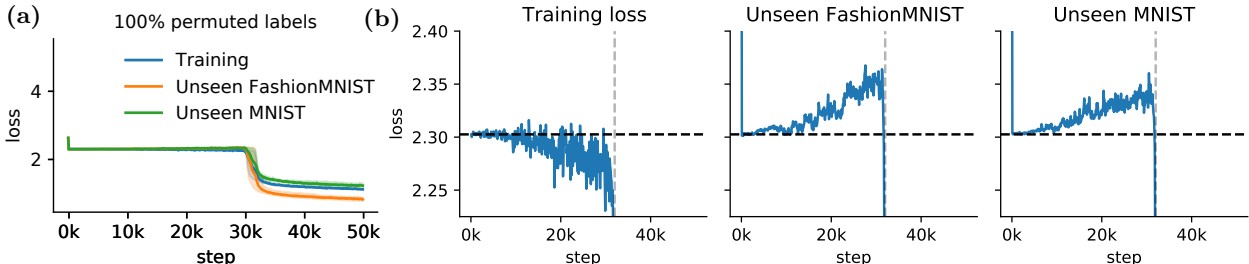

Figure 7: **Meta-training dynamics often involve an extended period where GPICL's performance is stuck on a plateau. (a)** Meta-loss vs. meta-training step, for a uniform distribution over meta-training tasks. Training tasks are generated by random transformations of FashionMNIST. **(b)** A zoomed in view of the plateau. The loss only decreases slightly and the model memorize small biases in the training data (decreasing generalization) before the loss drops sharply.

**What corresponds to state (memory) in various architectures?** Memory $N_S$ in the context of recurrent neural networks corresponds to the hidden state or context vector of size $N_H$, thus $N_S \in \mathcal{O}(N_H)$. More generally, we can describe the state as the information bottleneck that the sequence has to pass through before making predictions. In the context of learning-to-learn, this state has to hold information about everything that has been learned so far. Standard learning algorithms such as neural networks trained via SGD would have a state that corresponds to the neural network parameters, iteratively updated via SGD. In transformers, self-attention allows for a particularly large state of $N_S \in \mathcal{O}(N_K N_L N_T)$ where $N_K$ is the size of key, value, and query, $N_L$ is the number of layers, and $N_T$ is the length of the sequence. In addition to Figure 6, Figure 13 show meta-test performance on more tasks and datasets.

**Insight 4: Large state is more crucial than parameter count** This suggests that the model size in terms of parameter count plays a smaller role in the setting of learning-to-learn and Transformers have benefited in particular from an increase in state size by self-attention. Beyond learning-to-learn, this likely applies to other tasks that rely on storing large amounts of sequence-specific information.

### 4.3 Challenges in Meta-Optimization

Meta-optimization is known to be challenging. Meta gradients (Finn et al., 2017; Xu et al., 2018; Bechtle et al., 2021) and works with parameter sharing or weight updates in their architecture (Kirsch & Schmidhuber, 2020; Pedersen & Risi, 2021; Risi, 2021) observed various difficulties: Slower convergence, local minima, unstable training, or loss plateaus at the beginning of training (see Appendix Figure 21). We show that some of these problems also occur with black-box models and propose effective interventions.

**Loss plateaus when meta-learning with black-box models** By training across a large number of randomly transformed tasks, memorizing any task-specific information is difficult. Instead, the model is forced to find solutions that are directly learning. We observe that this results in (meta-)loss plateaus during meta-training where the loss only decreases slightly for long periods of time (Figure 7a). Only after a large number of steps (here around 35 thousand) does a drop in loss occur. In the loss plateau, the generalization loss increases on unseen tasks from both the same and a different base dataset (Figure 7b). This suggests that being able to first memorize slightly enables the following learning-to-learn phase. Furthermore, we observe that all gradients have a very small norm with exception of the last layer (Appendix Figure 17).

**Intervention 1: Increasing the batch size** High variance gradients appear to be one reason training trajectories become trapped on the loss plateau (see Appendix Figures 15, 16). This suggests increasing the meta-batch size as a straightforward solution. When plotting various batch sizes against numbers of tasks we obtain three kinds of solutions at the end of meta-training (Figure 8a): (1) Solutions that generalize and learn, (2) Solutions that memorize, and (3) Solutions that are still in the loss plateau (due to maximum of 50 thousand optimization steps). The larger the batch size, the more tasks we can train on without getting stuck in a loss plateau. When plotting the length of the loss plateau against the task batch size (Figure 8b) we observe a power-law relationship with increasing batch sizes decreasing the plateau length. At the same

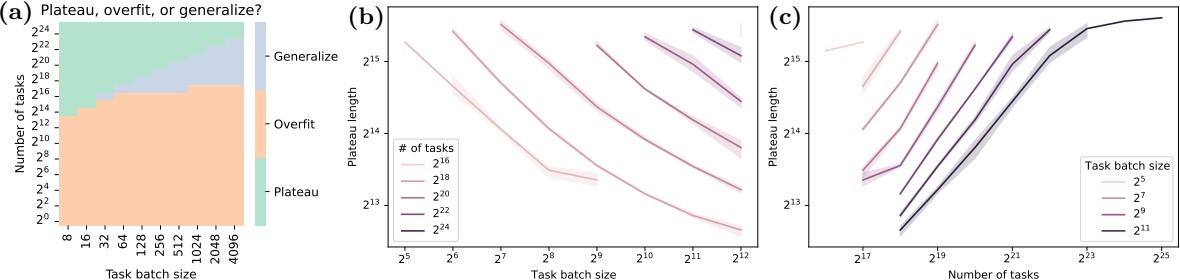

Figure 8: **Whether GPICL memorizes, generalizes, or remains trapped on a meta-loss plateau depends on the number of meta-training tasks, and the meta-training batch size. (a)** A phase diagram showing GPICL's behavior at the end of meta-training (50k steps). Solutions either memorize, generalize and learn, or remain in the loss plateau. With additional training steps, configurations in the plateau might eventually transition to memorization or generalization. Generalization only occurs with large enough batch sizes and sufficient, but not too many, tasks. **(b)** This behavior is explained by the plateau length decreasing with the increasing batch sizes (reducing the noise contribution), and **(c)** increasing with larger numbers of tasks.

time, the batch size also increases the number of total tasks seen in the plateau (Appendix Figure 18). Thus, this intervention relies on parallelizability. An increase in the number of tasks also increases the plateau length (Figure 8c), possibly due to a larger number of tasks inhibiting the initial memorization phase.

**Intervention 2: Changes in the meta-optimizer** Given that many gradients in the loss plateau have very small norm, Adam would rescale those element-wise, potentially alleviating the issue. In practice, we observe that the gradients are so small that the $\epsilon$ in Adam's gradient-rescaling denominator (for numerical stability) limits the upscaling of small gradients. Using smaller $\epsilon$ results in more than halving the plateau length. Alternatively, discarding the magnitude of the gradient entirely by applying the sign operator to an exponential moving average of the gradient (replacing Adam's approximate magnitude normalization with direct magnitude normalization) has a similar effect while also increasing the numerical stability over Adam with small $\epsilon$ (Appendix Figure 19).

**Intervention 3: Biasing the data distribution / Curricula** GPICL mainly relies on the data distribution for learning-to-learn. This enables a different kind of intervention: Biasing the data distribution. The approach is inspired by the observation that before leaving the loss plateau the model mem-

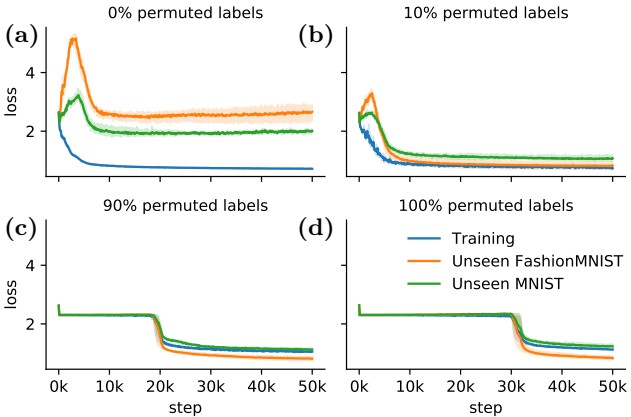

Figure 9: **Biasing the training distribution is an effective intervention which prevents a meta-loss plateau.** A uniform distribution over tasks leads to a long plateau **(d)**, while increasing the training fraction that corresponds to a single task reduces the plateau **(abc)**.

orizes biases in the data. Instead of sampling label permutations uniformly, we bias towards a specific permutation by using a fixed permutation for a fraction of each batch. This completely eliminates the loss plateau, enabling a smooth path from memorizing to learning (Figure 9). Surprisingly, even when heavily biasing the distribution, memorization is followed by generalization. This biased data distribution can be viewed as a curriculum, solving an easier problem first that enables the subsequent harder learning-to-learn. Further investigation is required to understand how this transition occurs. This may be connected to grokking (Power et al., 2022) which we investigate in Appendix A.5. We hypothesize that many natural data distributions—including language—contain such sub-tasks that are easy to memorize followed by generalization.

### 4.4 Domain-specific and general-purpose learning

We demonstrated the feasibility of meta-learning in-context learning algorithms that are general-purpose. An even more useful learning algorithm would be capable of both generalizing, as well as leveraging domain-specific information for learning when it is available. This would allow for considerably more efficient in-context learning, scaling to more difficult datasets without very long input sequences. Toward this goal, we investigate a simple scheme that leverages pre-trained neural networks as features to learn upon. This could be from an unsupervised learner or a frozen large language model (Radford et al., 2021; Tsimpoukelli et al., 2021). Here, we first project the inputs $\bar{x}_i$ of a base-dataset $\bar{D}$ into some latent space using a pre-trained network, and then proceed with meta-training and meta-testing as before, randomly projecting these alternative features. For the pre-trained network, we use a ResNet trained on ImageNet and remove its final layer. In Figure 10 we have meta-trained GPICL on MNIST either with the randomly transformed raw inputs or randomly transformed embedded features. At meta-test-time the learning algorithm generalizes to a wide range of datasets, measured by the meta-test accuracy of the 100th example. At the same time, the pre-trained ImageNet helps to accelerate learning on datasets that have a matching domain, such as CIFAR10. We observe that with only 100 examples, the learning algorithm

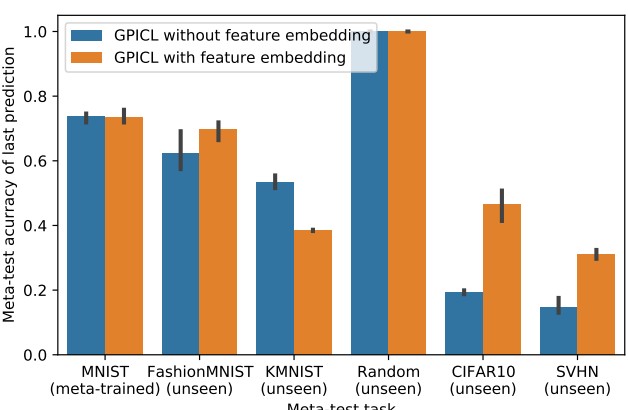

Figure 10: **Using pre-trained networks allows leveraging domain-specific knowledge while still generalizing to other datasets** GPICL is meta-trained on MNIST either with the randomly transformed raw inputs or randomly transformed pre-trained features. Pre-training helps to accelerate meta-test-time in-context learning on datasets that have a matching domain, such as CIFAR10. With only 100 examples, the learning algorithm can achieve about 45% accuracy on CIFAR10. The learning algorithms still generalize to a wide range of datasets. Error bars are 95% confidence intervals of the mean across meta-training runs.

meta-trained on MNIST, can achieve about 45% accuracy on CIFAR10.

## 5 Related work

**Meta-learning: Inductive biases and general-purpose learning algorithms** Meta-learning approaches exist with a wide range of inductive biases, usually inspired by existing human-engineered learning algorithms. Some methods pre-wire the entire learning algorithm (Finn et al., 2017), pre-wire backpropagation and the structure of a gradient-based optimizer (Andrychowicz et al., 2016; Metz et al., 2019; 2020a), or learn the loss function (Houthooft et al., 2018; Kirsch et al., 2019; Bechtle et al., 2021).

Many methods search over hyper-parameters that alter existing learning algorithms (Xu et al., 2018; Metz et al., 2020b; Chen et al., 2022). Fast weight programmers or hyper-networks update the weights of the same or another neural network (Schmidhuber, 1992; 1993a; Ha et al., 2017; Irie et al., 2021; Sandler et al., 2021; Kirsch & Schmidhuber, 2022; Zhmoginov et al., 2022), frequently with various symmetries. There has been growing interest in meta-learning more general-purpose learning algorithms. Such learning algorithms aim to be general and reusable like other human-engineered algorithms (e.g. gradient descent). The improved generality of the discovered learning algorithm has been achieved by introducing inductive bias, such as by bottlenecking the architecture or by hiding information, encouraging learning over memorization. Methods include enforcing learning rules to use gradients (Metz et al., 2019; Kirsch et al., 2019; Oh et al., 2020), symbolic graphs (Real et al., 2020; Co-Reyes et al., 2021), or parameter sharing and symmetries (Kirsch & Schmidhuber, 2020; Kirsch et al., 2021). Parameter sharing and symmetries have additionally been discussed in the context of self-organization (Tang & Ha, 2021; Risi, 2021; Pedersen & Risi, 2022).

**In-context learning with black-box models** Black-box neural networks can learn-to-learn purely in their activations (in-context) with little architectural and algorithmic bias (Hochreiter et al., 2001; Wang et al., 2016; Duan et al., 2016; Santoro et al., 2016; Garnelo et al., 2018). This requires a feedback or demonstration signal in the inputs that allows for learning such as the reward in reinforcement learning or label in supervised learning (Schmidhuber, 1993b). While a frequently used architecture is the LSTM (Hochreiter & Schmidhuber, 1997; Gers et al., 2000), this mechanism has also seen substantial recent attention in Transformer models (Brown et al., 2020; Chan et al., 2022) under the name of in-context learning. In large language models (LLMs) demonstrations of a task in the input help solving language-based tasks at inference (meta-test) time (Brown et al., 2020). This few-shot learning ability has been attributed to the data-distributional properties of text corpora (Chan et al., 2022). In-context learning has also been interpreted from a Bayesian inference perspective (Ortega et al., 2019; Mikulik et al., 2020; Nguyen & Grover, 2022; Müller et al., 2022). Our method GPICL is in the class of these black-box in-context learners. The number of model parameters has been at the core of scaling up LLMs to unlock greater capabilities and have been formulated in scaling laws (Kaplan et al., 2020; Hoffmann et al., 2022). Our empirical study suggests that for learning-to-learn, the amount of memory (model state) is even more predictive of in-context learning capabilities than parameter count.

**General-purpose in-context learning** While in-context learning has been demonstrated with black-box models, little investigation of general-purpose meta-learning with these models has been undertaken. Generalization in LLMs has previously been studied with regards to reasoning and systematicity (Csordás et al., 2021; Delétang et al., 2022; Wei et al., 2022; Zhou et al., 2022; Anil et al., 2022). In this work we focus on meta-generalization instead, the extent to which in-context learning algorithms generalize. In contrast to previous methods, GPICL implements *general-purpose* learning algorithms. Independently, Garg et al. (2022) recently studied generalization on synthetic functions compared to our augmented datasets. VSML (Kirsch & Schmidhuber, 2020) also implements in-context learning with black-box LSTMs, but makes use of parameter-sharing to aid generalization. PFNs (Müller et al., 2022) demonstrated learning to learn on small tabular datasets when meta-training on synthetically generated problems. Experiments on more complex classification settings such as Omniglot relied on fine-tuning. In comparison, our method investigated meta-generalization of learning algorithms directly to datasets such as MNIST, Fashion MNIST, and CIFAR10 while studying fundamental questions about the conditions necessary for such generalization. TabPFNs (Hollmann et al., 2022) extend PFNs to larger tabular datasets.

# 6 Discussion and Conclusion

By generating tasks from existing datasets, we demonstrated that black-box models such as Transformers can meta-learn general-purpose in-context learning algorithms (GPICL). We observed that learning-to-learn arises in the regime of large models and large numbers of tasks with several phase transitions from task memorization, to task identification, to general learning. The size of the memory or model state significantly determines how well any architecture can learn how to learn across various neural network architectures. We identified difficulties in meta-optimization and proposed interventions in terms of optimizers, hyperparameters, and a biased data distribution acting as a curriculum. We demonstrated that in-context learning algorithms can also be trained to combine domain-specific learning and general-purpose learning. We believe our findings open up new possibilities of data-driven general-purpose meta-learning with minimal inductive bias, including generalization improvements of in-context learning in large language models (LLMs).

An important subject of future work is the exploration of task generation beyond random projections, such as augmentation techniques for LLM training corpora or generation of tasks from scratch. A current limitation is the applicability of the discovered learning algorithms to arbitrary input and output sizes beyond random projections. Appropriate tokenization to unified representations may solve this (Chowdhery et al., 2022). Furthermore, learning algorithms often process millions of inputs before outputting the final model. In the current black-box setting, this is still difficult to achieve and it requires new advances for in context length of sequence models. Recurrency-based models may suffer from accumulating errors, whereas Transformer's computational complexity grows quadratically in sequence length.

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

# A  Appendix

## A.1  Summary of Insights

**Insight 1: It is possible to learn-to-learn with black-box models**  Effective in-context learning algorithms can be realized using black-box models with few inductive biases, given sufficient meta-training task diversity and large enough model sizes. To transition to the learning-to-learn regime, we needed at least $2^{13} = 8192$ tasks.

**Insight 2: Simple data augmentations are effective for general learning-to-learn**  The generality of the discovered learning algorithm can be controlled via the data distribution. Even when large task distributions are not (yet) naturally available, simple augmentations that promote permutation and scale invariance are effective.

**Insight 3: The meta-learned behavior has phase transitions**  When increasing the number of tasks, the meta-learned behavior transitions from task memorization, to task identification, to general learning-to-learn. The last transition is discrete, with two unique clusters.

**Insight 4: Large state is more crucial than parameter count**  The specific inductive biases of each architecture matter to a smaller degree. The driving factor behind their ability to learn how to learn is the size of their state. Furthermore, this suggests that the model size in terms of numbers of parameters plays a smaller role in the setting of learning-to-learn and Transformers have benefited in particular from an increase in state size by self-attention. In non-meta-learning sequence tasks parameter count is thought to be the performance bottleneck (Collins et al., 2016). Beyond learning-to-learn, this likely applies to other tasks that rely on processing and storing large amounts of sequence-specific information.

## A.2  Limitations

**Varying input and output sizes**  Compared to many previous works in meta-learning (Andrychowicz et al., 2016; Finn et al., 2017; Kirsch & Schmidhuber, 2020), the discovered learning algorithms are only applicable to an arbitrary input and output size by using random projections. This may make it more difficult to apply the learning algorithm to a new, unseen problem. This problem also applies to Transformers applied to multiple tasks and modalities. Related work has solved this problem by tokenizing inputs to compatible, unified representations (Chowdhery et al., 2022). We expect these techniques or others to be useful in the learning-to-learn context too.

**Processing large datasets**  Learning algorithms often process millions of inputs before outputting the final model. In the black-box setting, this is still difficult to achieve. Recurrency-based models usually suffer from accumulating errors, whereas Transformers computational complexity grows quadratically in the sequence length. Additional work is required to build models capable of processing and being trained on long sequences. Alternatively, parallel processing, similar to batching in learning algorithms, may be a useful building block.

## A.3  Architectural Details and Hyper-parameters

**Transformer details**  By default, all Transformers have a key, value, and query size of 32, 8 heads, and 4 layers, and model size of $N_M = 256$. The model size defines the dimensionality of each token, and the MLP between layers scales this size up to a hidden representation of $4 \times N_M$ where $N_M$ corresponds to the model size.

**Outer-product LSTM**  We slightly modify an LSTM by replacing the context state with an outer-product update and inner-product read-out.

```
x_and_h = jnp.concatenate([inputs, prev_state.hidden], axis=-1)

gated = hk.Linear(8 * size * self.num_heads)(x_and_h)
gated = gated.reshape((batch_size, self.num_heads, 8 * size))
gated = checkpoint_name(gated, 'gated')
```

```
# i = input, g = cell_gate, f = forget_gate,
# q = query, o = output_gate
sizes = (3 * size, 3 * size, size, size)
indices = np.cumsum(sizes[:-1])
k1, k2, q, o = jnp.split(gated, indices, axis=-1)
scale = jax.nn.softplus(
    hk.get_parameter('key_scale', shape=(), dtype=k1.dtype,
                     init=jnp.zeros))
i, g, f = jnp.einsum('bhki,bhkj->kbhij',
                     jax.nn.tanh(split_axis(k1, (3, size))) * scale,
                     jax.nn.tanh(split_axis(k2, (3, size))))
f = jax.nn.sigmoid(f + 1)  # Forget bias
c = f * prev_state.cell + jax.nn.sigmoid(i) * g
read = jnp.einsum('bhij,bhi->bhj', c, q)
h = hk.Flatten()(jax.nn.sigmoid(o) * jnp.tanh(read))
```

**VSML**  We use a version of VSML with a single layer and self-messages (Kirsch et al., 2021) of size 8. Each LSTM has a hidden size of 16. For each LSTM update we use two micro-ticks. We train on $2^{25}$ tasks with a 90% biased permutation distribution. The task batch size is 8. All images are scaled to a size of $32 \times 32 \times 3$

**VSML without symmetries**  Before activations are fed to a standard instantiation of VSML, all inputs are projected using a learnable linear projection. Logits are generated using another linear projection, followed by a softmax. We use a version of VSML with a single layer and self-messages (Kirsch et al., 2021) of size 8. The LSTMs are on a grid of $k \times k$ LSTMs, where $k \in \{1, 2, 4, 8, 16, 24\}$. Each LSTM has a hidden size of 64. For each LSTM update we use two micro-ticks. We train on $2^{25}$ tasks with a 90% biased permutation distribution. The task batch size is 128. All images are scaled to a size of $14 \times 14$.

**LSTM**  For the results in Table 2, we used a hidden size of 256 and $10^5$ optimization steps. Larger hidden sizes were harder to optimize. We train on $2^{25}$ tasks with a 90% biased permutation distribution. The task batch size is 128. All images are scaled to a size of $32 \times 32 \times 3$

### A.4 Experimental Details

Most experiments can be run on a single GPU, some require 16 GPUs due to sequence length and large batch sizes, with sufficient GPU memory (around 16 GB each). Some experiments, such as Figure 2, require up to 1000 runs of that kind to produce the final heat-map.

**Input normalization**  Each dataset is z-normalized by its mean and standard deviation across all examples and pixels.

**Number of seeds and shading**  If not noted otherwise, line plots use 8 seeds for meta-training and at least 512 seeds for meta-testing. Shading indicates 95% confidence intervals.

**Random dataset**  To test the meta-learned learning algorithms on a synthetically generated problem, we generate classification datasets of 10 datapoints where the input $x \in \mathbb{R}^{32 \times 32 \times 3}$ is drawn from a uniform distribution between 0 and 1. For each datapoint, labels $y$ are drawn from a uniform categorical distribution of 10 classes.

**Figure 2**  The MLP has two hidden layers of varying size with relu activations. The Transformer has the default parameters as defined above.

**Figure 3**  We use a transformer model with a model size of 256. We train on $2^{25}$ tasks with a 90% biased permutation distribution. The task batch size is 128. All images are scaled to a size of $32 \times 32 \times 3$ Inputs are z-normalized across the dataset and all input dimensions.

**Table 2**  The SGD baseline was obtained by sweeping over learning rates from $10^{-4}$ to 0.5, optimizers SGD, Adam and Adam with weight decay, one or two layers, and hidden sizes of 32, 64, or 128 on MNIST. The best configuration (most sample efficient) corresponds to a learning rate of $10^{-3}$, Adam, and no hidden layers.

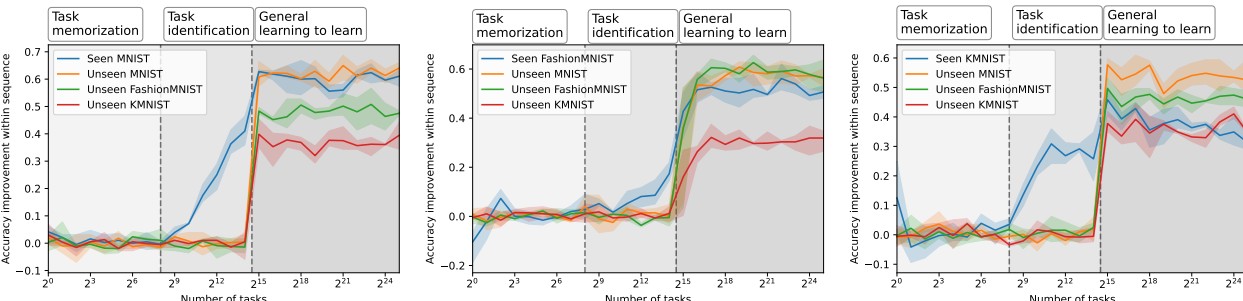

Figure 11: **Transformers exhibit three different phases in terms of meta-learned behavior on various meta training datasets.** (1) When training on a small number of tasks, tasks are memorized. (2) Tasks from the training distribution are identified, which is evident as a within-sequence increase of performance. (3) When training across many tasks, we discover a learning algorithm that generalizes to unseen tasks and unseen datasets.

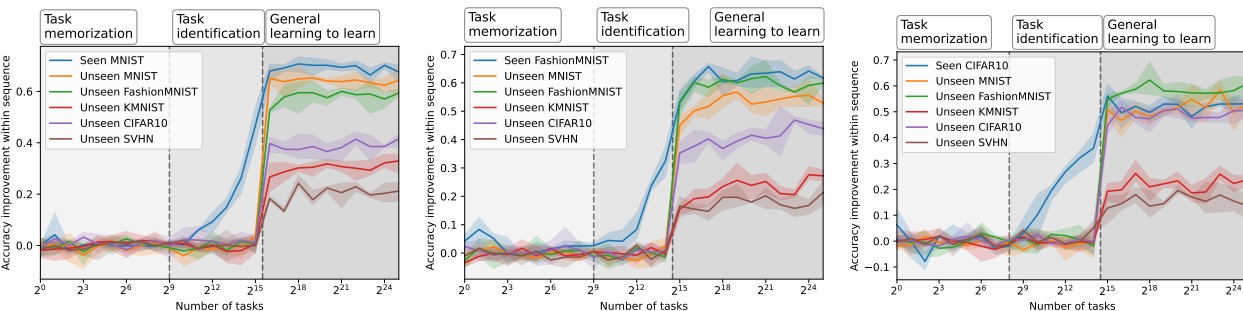

Figure 12: **The phase transitions also happen when using the embeddings from Section 4.4.** This enables faster learning on datasets such as CIFAR10 with only 100 training examples while still generalizing to various datasets.

SGD performs updates online on each one out of the 100 data points. MAML is equivalent to SGD up to the difference that we meta-train the weight initialization according to Equation 2 where $\theta$ are the initial parameters of the classifier that is then updated using SGD at meta-test time. All black-box approaches do not use gradient descent at meta-test time. All meta-learning approaches where meta-trained and tuned via grid search on MNIST.

**Figure 4** Input normalization is disabled.

**Figure 6** The Transformer uses a task batch size of 512.

**Figure 7** Trained on $2^{16}$ tasks generated from FashionMNIST with labels fully permuted.

**Figure 8** Trained on $2^{16}$ tasks generated from FashionMNIST with labels fully permuted.

**Figure 9** Trained on $2^{16}$ tasks generated from FashionMNIST with label permutations varied.

**Figure 5** We trained a Transformer with model size 64 and 32 seeds for each number-of-tasks-configuration.

## A.5 Additional Experiments

**Phase transitions on other meta training datasets** In Figure 2 and Figure 4 we observe a fairly discrete transition between task identification and general learning-to-learn as a function of the number of tasks. We show these phase transitions on more meta training datasets in Figure 11. When using ImageNet embeddings as discussed in Section 4.4, we observe similar phase transitions also on CIFAR10 and other datasets as shown in Figure 12.

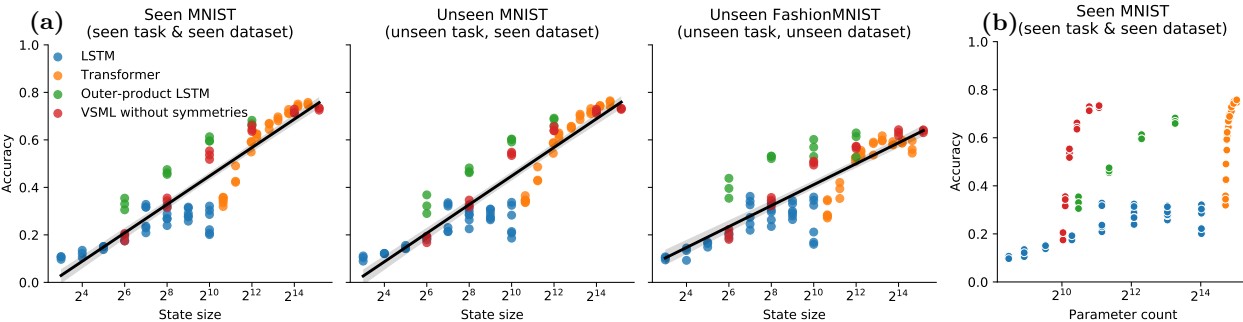

Figure 13: **The state size (accessible memory) of an architecture most strongly predicts its performance as a general-purpose learning algorithm.** **(a)** A large state is crucial for learning-to-learn to emerge. **(b)** The parameter count correlates less well with learning capabilities.

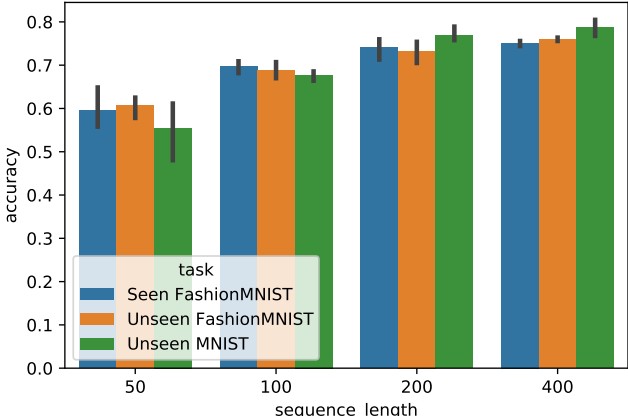

Figure 14: Increasing the sequence length during meta-training and meta-testing improves the predictive performance of the final query in the sequence. Error bars indicate 95% confidence intervals.

**Large State is Crucial for Learning** We show that for learning-to-learn the size of the memory $N_S$ at meta-test time (or state more generally) is particularly important in order to be able to store learning progress. We test this by training several architectures with various $N_S$ in our meta-learning setting. In addition to Figure 6, Figure 13 show meta-test performance on more tasks and datasets.

**Sequence length** In all experiments of the main paper we have meta-trained on a sequence length (number of examples) of 100. This is a small training dataset compared to many human-engineered learning algorithms. In general, as long as the learning algorithm does not overfit the training data, more examples should increase the predictive performance. In Figure 14 we investigate how our model scales to longer sequence lengths. We observe that the final accuracy of the last query in the sequence consistently increases with longer sequences. The generalization to longer sequences than those seen during meta-training is another important direction for future work.

**Gradient and update statistics** To better understand the properties of the loss plateau, we visualize different statistics of the gradients, optimizer, and updates. In Figure 15, we track the exponential moving average statistics of Adam before the loss plateau and after (dashed vertical line). In Figure 16 we investigate how gradients differ between settings with a plateau and settings with a biased distribution where the plateau is avoided. We plot the cosine similarity between consecutive optimization steps, the gradient L2-norm, and the similarity and norm of the weight updates after normalization with Adam. The statistics are plotted cumulatively or smoothed with a Gaussian filter for better readability. The gradient and update cosine similarity differ only marginally between cases with a plateau and cases without. We observe that the gradient L2-norm in the plateau is half as big as in the biased distribution case, although the updates that

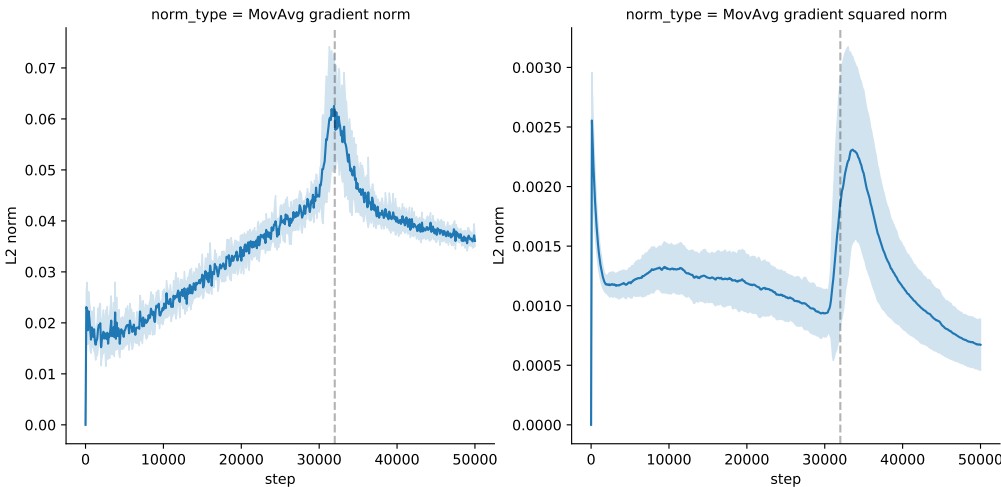

Figure 15: L2-norms of the gradient and squared gradient exponential moving average in Adam. The dashed line corresponds to the loss drop at the end of the loss plateau.

Adam applies are going towards zero. This also results in not moving far from parameter initialization when in the plateau. We hypothesize this has to do with varying gradient norms when looking at individual parameter tensors (Figure 17). We observe that the gradients have a small norm for most tensors, except for the last layer.

**Batch size and number of tasks influence on plateau length** Instead of looking at the plateau length in terms of the number of steps (Figure 8), we may also be concerned with the total number of tasks seen within the plateau. This is relevant in particular when the task batch is not processed fully in parallel but gradients are accumulated. Figure 18 shows the same figure but with the number of tasks in the plateau on the y-axis instead. It can be observed that larger batch-sizes actually increase the data requirement to leave the plateau, despite decreasing the plateau in terms of the number of optimization steps. Similarly, a larger task training distribution requires a larger number of tasks to be seen within the plateau.

**Adjusting Adam's $\epsilon$ or changing the optimizer** As discussed in the main paper and visualized in Figure 19b, decreasing $\epsilon$ significantly shortens the plateau. This is due to the rescaling of very small gradient magnitudes being limited by $\epsilon$. At the same time it incurs some instability. Directly normalizing the gradient by applying the sign function element-wise (Figure 19a) to the exponential gradient average shortens the plateau even further.

**When memorization happens, can we elicit grokking?** In Figure 8a we have seen that an insufficiently large task distribution can lead to memorization instead of general learning-to-learn. At the same time, Figure 9 showed that biasing the data distribution is helpful to avoid loss plateaus. Power et al. (2022) observed a phenomenon which they called "grokking" in which even after having converged in terms of training loss, test loss may suddenly decrease. Large amounts of regularization, like weight decay with a coefficient of 1.0 were found to facilitate this behavior. Is grokking connected to the optimization behavior we observe, and if so, do similar interventions help in our setting? We look in particular at the boundary of memorization and generalization ($2^{14} = 16384$) where doubling the number of tasks a few more times would lead to generalization. Figure 20 shows three task settings, $2^{10}, 2^{14}, 2^{16}$, and three different weight decay coefficients, $0.01, 0.1, 1.0$. The setting of $2^{16}$ tasks shows generalization by default and only serves as a baseline for the weight decay coefficient analysis. In the cases of memorization due to too few tasks, we have not been able to produce grokking behavior.

**Optimization difficulties in VSML** Previous work has observed several optimization difficulties: Slower convergence, local minima, unstable training, or loss plateaus at the beginning of training. Figure 21 shows some of these difficulties in the context of VSML (Kirsch & Schmidhuber, 2020). Because VSML has permutation invariance and parameter sharing built into the architecture as an inductive bias, changing the

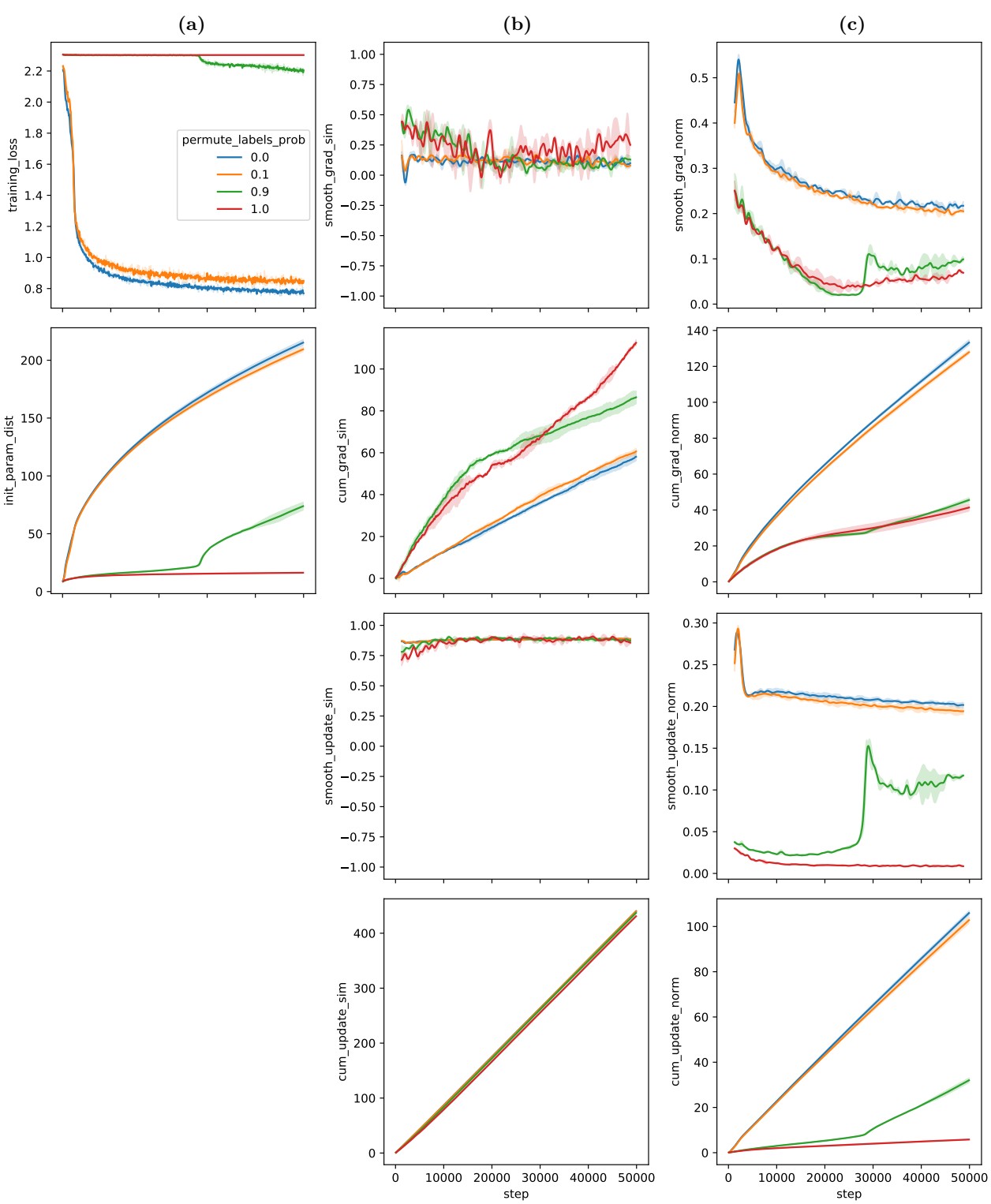

Figure 16: Gradient and Adam update statistics for differently biased data distributions. **(a)** Plateaus in the loss are influenced by the bias in the data distribution. Plateaus result in moving away slowly from the parameter initialization. **(b)** The cosine similarity of both gradients and updates in consecutive steps is only marginally different with or without a loss plateau. **(c)** While the gradient norm is about half as big when a plateau exists, the updates are going towards zero.

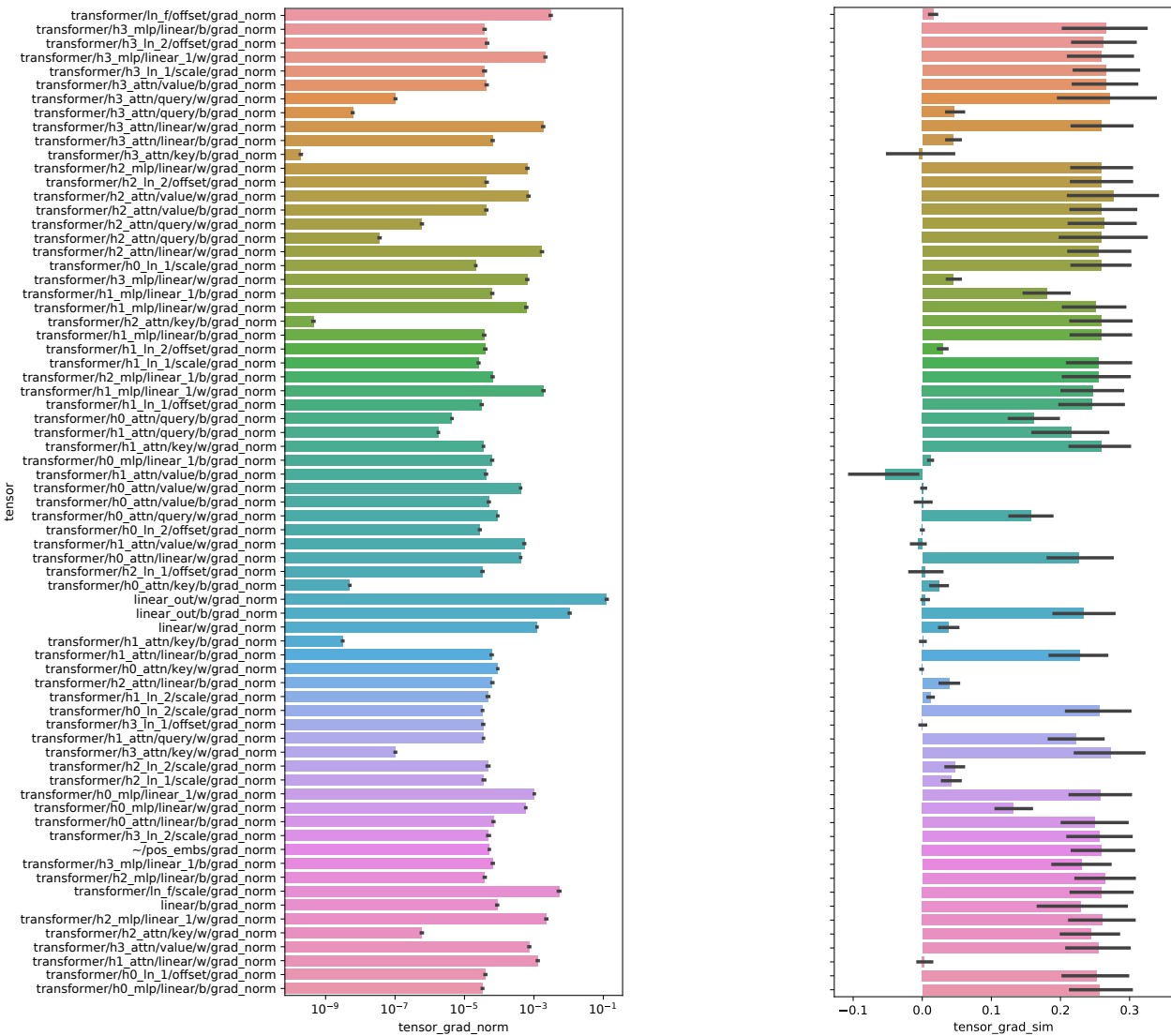

Figure 17: Gradient L2 norms (left) and gradient cosine similarity for consecutive optimization steps (right) for different parameter tensors. The last (output) layer has the largest gradients. Most other gradients are small.

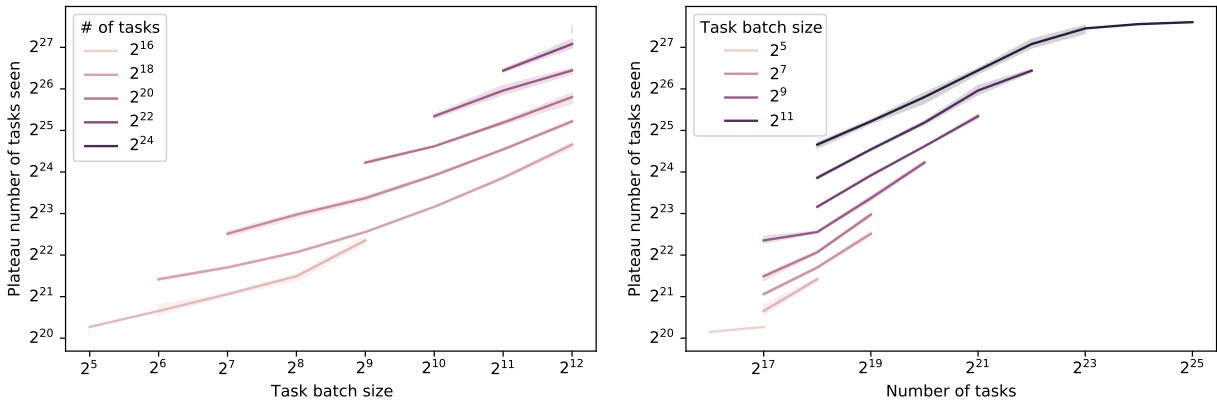

Figure 18: Instead of plotting the loss plateau length in terms of optimization steps, we look at the total number of tasks seen within the plateau as a function of the task batch size and the number of tasks in the training distribution. An increase in the task batch size leads to more tasks to be processed to leave the plateau.

Figure 19: **(a)** When replacing Adam with a sign normalization of the gradient or **(b)** reducing $\epsilon$ the plateau length is significantly shorter.

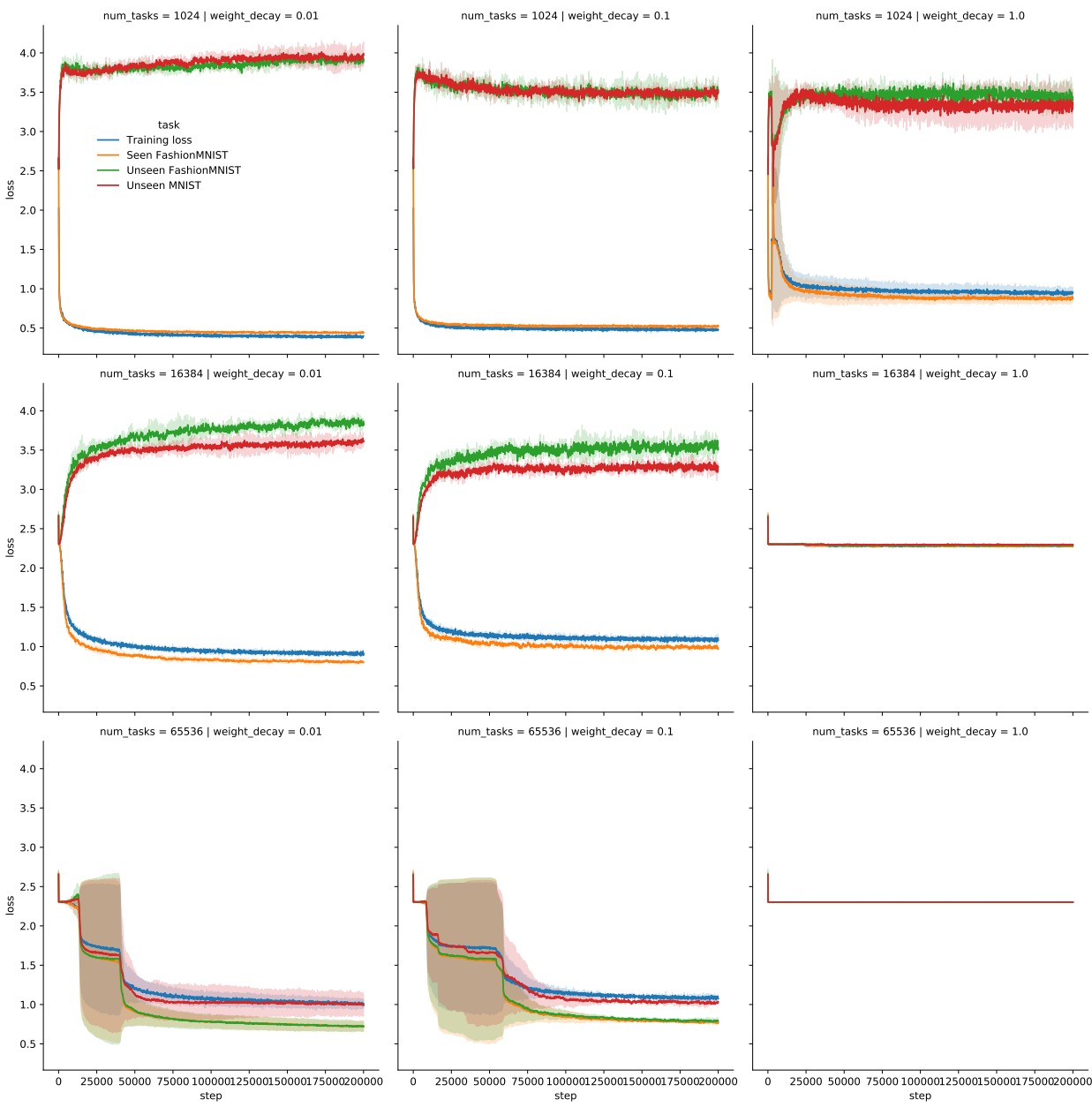

Figure 20: We investigate whether grokking as defined in Power et al. (2022) can be produced when we observe memorization on a smaller numbers of tasks. This would correspond to the test loss decreasing long after the training loss has converged. We have not been able to elicit this behavior when looking at different numbers of tasks and weight decay coefficients.

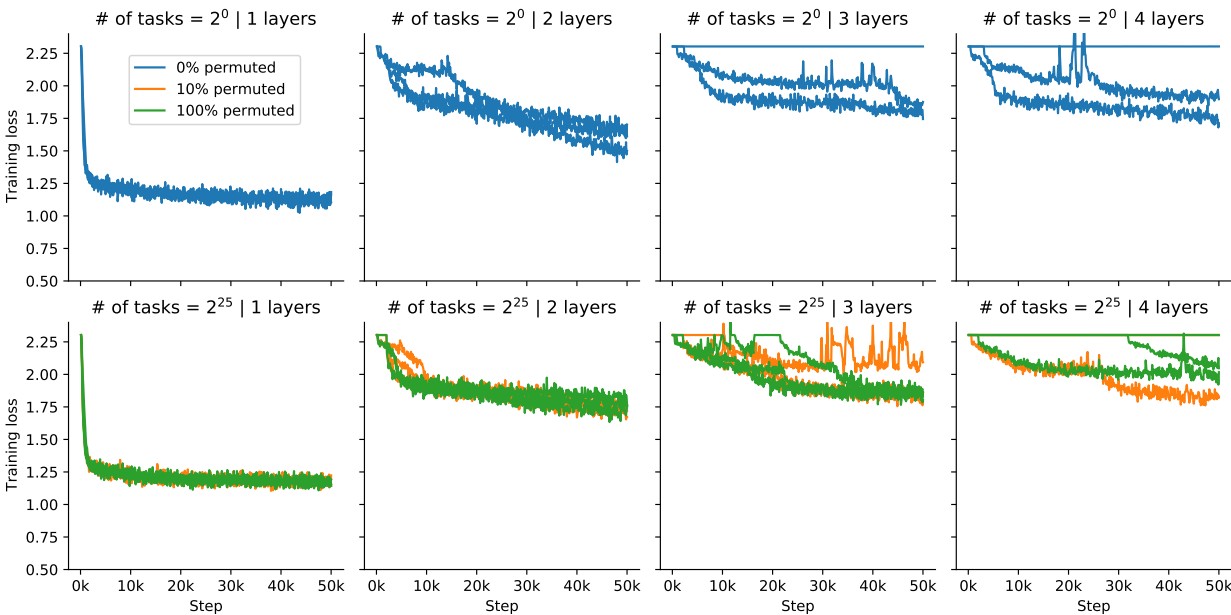

Figure 21: Loss plateaus and slow convergence with deeper variants of VSML.

number of tasks has only a small effect. We observe that in particular deeper architectures make meta-optimization more difficult.

