# OpenReview forum: "General-Purpose In-Context Learning by Meta-Learning Transformers"
_TMLR — Rejected by TMLR_

### Review · Reviewer_5w98 · 2023-07-06

**Summary Of Contributions:**

The paper presents a formulation for general in-context learning using transformers. The key contribution is using random projections to generate sufficiently diverse tasks from limited datasets and induces the transformers to generalize. Several practical techniques for learning such a transformer is presented, such as adjusting the optimizer used and changing the batch size.

Detailed empirical results are presented to explain/investigate the behaviors of the meta-learning transformer, including observations such as phase transition, and the importance of state size for generalization.

**Audience:**

Yes

**Broader Impact Concerns:**

N.A.

**Claims And Evidence:**

No

**Requested Changes:**

Necessary changes:

1 . Many claimed insights are based on experiments with fairly homogeneous data from relatively simple vision datasets. More extensive experiments are required to support the claimed insights, such as the phase transition and the role of state sizes. For instance, it may be conceivable that MNIST-style datasets are simple enough that adding parameter counts doesn't improve the performance. However, it is unclear that whether this would remain true for other datasets.

Inclusion of non-vision datasets such as tabular data would be a nice addition to test the general in-context learning ability of transformers, alongside vision data.

2 . As discussed in the Weaknesses above,  it is unclear that if the claimed phase transition is a genuine phenomenon. More rigorous explanation or experiments are needed to support this claim. The authors should also contextualize the significance of such phase transition.

3 . As it stands, the model performance is quite poor (especially on CIFAR and SVHN datasets). The results appear rather preliminary and it is difficult to imagine how the proposed approach could be used as is. In particular, the random projection strategy appears to destroy the spatial structures for visual input and the general in-context learner struggles to learn meaningful meta-knowledge across tasks (e.g., compared to domain-specific embedding in Fig. 10). Even with the ResNet embedding, the generalization performance still appears relatively weak. Changes o the meta-learning regime should be considered to improve model performance.

**Strengths And Weaknesses:**

Strength:

 1 . Very Interesting and useful observations/discussion on how to train the meta-learning transformers, (e.g., the loss plateau, the behavior of the optimizer and leveraging curricula to speed up training).

 2 . The presentation is mostly clear and easy to follow.

Weakness:
1 . As the authors pointed out at conclusion, the scope of the empirical experiments appears fairly restricted, including only simple vision datasets. Thus it is unclear that the insights claimed in this paper could generalizes to more realistic scenarios. Even within the scope of the existing experiment, 45% accuracy on CIFAR dataset hardly indicates a robust meta-learner. In addition, the model as it stands could not handle input/output of arbitrary sizes.

2 . Some of the claimed insights appears overstretched. For instance, I'm unable to see a clear phase transition in Fig. 2. It appears more like a smooth change across the two axis. In addition, [1] recently argued that it is possible that emergent properties (similar to phase transition in this work) may be a artifact of the chosen metric of evaluation. I think it's therefore pertinent to further analyze/verify the phase transition presented. In particular, Fig 4. uses accuracy improvements within a sequence as the metric of choice. I find this metric unusual in the sense that it may accidentally penalize high-performing models: accuracy improvement would be zero for any model that classify everything correctly.

Further, the notion of task-identification is vague and not clearly defined: why is task-identification able to help with accuracy improvements within the sequence? It would appear that as long as the model could memorize a task, all samples within the task should be classified correctly. So how is this particular construct of task-identification phase significant?

[1] Schaeffer, R., Miranda, B., & Koyejo, S. (2023). Are emergent abilities of Large Language Models a mirage?. arXiv preprint arXiv:2304.15004.

---

> ### Author Response · Authors · 2023-08-14
> **Rebuttal (1 of 3)**
>
> Thank you for taking the time to review our paper and your insightful comments. We appreciate that you found our observations and discussions on training meta-learning transformers interesting and useful.
>
> Let us preface our response with some thoughts on why we chose our specific experimental setup.
> The objective of the paper is to investigate under what conditions black-box models such as Transformers learn to implement general-purpose learning algorithms.
> From this it follows that if such a general-purpose learning algorithm is implemented, sample efficiency is much lower compared to ‘few-shot learners’ that are trained on a certain task distribution with the expectation that it will only learn on / generalize to distributions similar to it.
> Another way to look at this is that in our setup very little ‘meta-knowledge’ can be extracted from the task distribution because this would ultimately impede generalization.
> Essentially, compared to most meta-learning setups we are studying the extreme of strong generalization but little exploitable domain-specific learning.
> To study this without creating overly complex task generators, we do so by generating random projections that create enough task diversity (but also destroy a lot of the structure).
> We don’t think our method should be used as-is but rather analyzes fundamental capabilities of black-box models as they may emerge in large language models (LLMs) with sufficient task diversity.
> We expect that future meta learning systems would operate at the combination of both extremes -- both learning domain-specific biases that allow for quick in-context learning as well as less efficient but more general learning algorithms. This requires task distributions that have both information that can be exploited as meta-knowledge but are also diverse enough to allow for very generalizable learning algorithms to be meta-learned.
>
> ### About model performance on more complex tasks
> Our model performance on more complex tasks does not match the performance of more domain-specific meta-learners. Strong performance on standard benchmarks was not the objective of this work. Instead, as discussed above, we focused on meta learning general learning algorithms that can learn across a wide range of tasks. To this end, we decided to select a simple task generation scheme based on random projections. Generating sufficient task diversity based on other augmentations or natural task distributions can be difficult and is an entire research area. Natural task distributions would potentially also introduce other biases to our analysis. Because of the general-purpose nature of the meta-learned learning algorithms, there is a fundamental limit of how good a classifier can be from just 100 examples. In Figure 14 of the appendix we showed that increasing the context length generally improves in-context learning performance. To perform well on complex datasets we would at least have to be able to attend to the entire dataset once, such as a context length of 60000 images of CIFAR10, which is expensive to do with Transformers, and is currently not practically feasible. Increasing the context length of Transformers and related architectures is an open research question. That said, we have experimented with several different smaller datasets and the results, e.g. in terms of the algorithmic transitions in the number of tasks, seem consistent (Figure 4, 11, 12). We believe that even without more complex datasets, we have discovered interesting phenomena that are worthy of publication.
> Regarding tabular datasets, in Table 2 we also observe strong meta-test performance on randomly generated datasets consisting of 10 datapoints (x, y) where x is a random uniform vector assigned to a random label y. This could of course be extended to other (natural) tabular datasets but we would not expect this to differ significantly from our existing results.

---

> > ### Author Response · Authors · 2023-08-14
> > **Rebuttal (2 of 3)**
> >
> > ### About phase transitions
> > One central question of the paper is under which conditions these general-purpose learning algorithms emerge. In our experiments we have observed several transitions in terms of the algorithmic solutions that are implemented. Two kinds of these transitions are shown in Figure 2, where the model starts generalizing with a sufficiently large model size and task distribution. We have looked at the transition in terms of the number of tasks in more detail (Figure 4). “We observe three phases: In the first phase, the model memorizes all tasks, resulting in no within-sequence performance improvement. In the second phase, it memorizes and learns to identify tasks, resulting in a within-sequence improvement confined to seen task instances. In the final and third phase, we observe a more general learning-to-learn, a performance improvement for unseen tasks, even different base datasets (here FashionMNIST)”.
> > We show accuracy improvement instead of absolute accuracy to investigate whether the model is learning at meta-test time (i.e. as a function of data observed within that episode). A model could overfit the task distribution and memorize the tasks perfectly (and that happens in the first phase, also see Figure R1 below) but, by construction of our task distribution, it has to learn how to learn to do well on unseen tasks.
> > Regarding your request to define task identification more clearly, here the model identifies which of the projections (tasks) observed during training it is seeing online, as opposed to the model having learned a general mechanism for handling new unseen tasks. In Figure 4 this is visible as a within-sequence accuracy improvement that is larger than zero only for the seen tasks. To add to this evidence, we can plot the same experiments not by their within-sequence improvement but simply plot the accuracy on the first inner step (no examples seen) and the last one: [Figure R1](https://ibb.co/qL8c93z)
> >
> > Regarding the nature of these transitions, in Figure 4 we see that the transition to task identification is slow and incremental over several orders of magnitudes, whereas the transition to general learning-to-learn happens more quickly on an exponential scale.
> > These transitions can be observed across several meta-training datasets, see Figures 11 and 12 in the appendix.
> > In some cases we found the second transition point to be non-smooth in the sense that training runs end up in one of two clusters in terms of the algorithmic solutions that are implemented (task identification or general learning to learn). This turns out to be true even when plotting the loss instead of the accuracy. In the paper, we have visualized that in Figure 5 where the training loss ends up in two distinct clusters when training with multiple seeds. What does this look like in terms of test losses? Based on the observed second transition in Figure 4, we have now plotted the test losses on seen and unseen datasets: For MNIST training in [Figure R2](https://ibb.co/x1XKDwR) and for KMNIST  training in [Figure R3](https://ibb.co/r4VqdTD).
> >
> >
> > In these plots, we observe that the generalization to unseen tasks does not smoothly increase but rather ends up in two clusters where more runs end up in the generalization cluster the larger the number of tasks. We believe this is an interesting phenomenon that creates opportunities for future research investigations.
> > We recently noticed that on some datasets such as FashionMNIST ([Figure R4](https://ibb.co/BrDkzTn)) this transition is more smooth in terms of test losses but still changes its ‘within sequence learning behavior’ in three phases as in Figure 11.
> >
> > In summary, we demonstrated that there exist transitions between different algorithmic solutions and analyzed how quickly they happen based on changes in task diversity. We further discovered that some of these transitions are highly non-smooth. We will clarify that in the paper.

---

> > > ### Author Response · Authors · 2023-08-14
> > > **Rebuttal (3 of 3)**
> > >
> > > ### About the role of state / memory size
> > > In section 4.2, we are claiming that across architectures a sufficiently large size of the state or memory is crucial to enable in-context learning. We provide evidence for this in Figure 6 where larger memory enables better in-context learning. Why do we believe that is the case? When task diversity is very high (here by generating tasks via random projections) a lot about the task needs to be learned from data instead of being stored in the model parameters and this information needs to be stored in the state or memory of the model. As pointed out earlier, we are looking at the strong generalization regime in this paper, and thus little meta knowledge can be extracted from the data distribution.
> > > We agree with the reviewer that it is likely that on different distributions, where there is more exploitable meta knowledge, a lot more can be stored in the parameters and thus makes the parameter count significant again. We don’t think this is a weakness of our analysis, but rather we present a clean way of highlighting the importance of memory for very generalizing in-context learners. That said, memory size will still remain relevant for all in-context learners. The more information needs to be learned / extracted from the context at meta-test time, the larger the memory needs to be. We essentially looked at one extreme here and we expect other task distributions to both require large numbers of parameters and large amounts of memory. We will clarify this in the paper.
> > >
> > > Thank you again for your review and we are looking forward to discussing any remaining concerns that you might have.

---

> > > > ### Comment · Reviewer_5w98 · 2023-08-18
> > > > **Reply to rebuttal**
> > > >
> > > > I thank the authors for the detailed and in-depth explanations. I have carefully gone through the explanations but many of my concerns remain.
> > > >
> > > > For the method itself, I appreciate the additional experiments/figures on phase transition. However, those are still non-linear/discontinuous metrics that may show up as phase transition unintentionally [1]. Therefore I think it is still important to explore this, as these insights are supposed to be the core contributions. Even if the phase transitions do exist, I think it could be better to simply focusing on before and after the transition, since charactering the transition itself is difficult with existing experiments.
> > > >
> > > > More generally, I still find the proposed meta-learning regime, in particular task generation via random projection, unwieldy. While I agree that it introduces task diversity and forces meta-learners to perform in-context learning, its present form doesn't scale to data of different modalities (without perhaps using some task/domain-specific feature extractors). It also destroys structural information (and thus some meta-knowledge) in structural data (e.g., vision or language data) considered in this work. It is difficult to imagine how the proposed regime could integrate with the existing meta-learning methods (the two extreme ends of the meta-learning in authors' conceptualization).
> > > >
> > > > I also share Reviewer onxL's concern that whether the insights could scale to larger and realistic datasets.
> > > >
> > > > I can't recommend acceptance and think the paper would benefit from significant revision.
> > > >
> > > > [1] Schaeffer, R., Miranda, B., & Koyejo, S. (2023). Are emergent abilities of Large Language Models a mirage?. arXiv preprint arXiv:2304.15004.

---

> > > > > ### Author Response · Authors · 2023-08-18
> > > > > **Rebuttal discussion**
> > > > >
> > > > > Thank you for your response and for taking the time to discuss this further.
> > > > >
> > > > > > For the method itself, I appreciate the additional experiments/figures on phase transition. However, those are still non-linear/discontinuous metrics that may show up as phase transition unintentionally [1]. Therefore I think it is still important to explore this, as these insights are supposed to be the core contributions. Even if the phase transitions do exist, I think it could be better to simply focusing on before and after the transition, since charactering the transition itself is difficult with existing experiments.
> > > > >
> > > > > We are happy to adjust our framing from ‘phase transitions’ to ‘algorithmic transitions’. The kind of solutions implemented in different task regimes are visible in Figures 4,11,12 (independent of the nature of these transitions) and are, from our viewpoint, the important insight. We think this change in framing can be done in a minor revision.
> > > > >
> > > > > > More generally, I still find the proposed meta-learning regime, in particular task generation via random projection, unwieldy. While I agree that it introduces task diversity and forces meta-learners to perform in-context learning, its present form doesn't scale to data of different modalities (without perhaps using some task/domain-specific feature extractors). It also destroys structural information (and thus some meta-knowledge) in structural data (e.g., vision or language data) considered in this work. It is difficult to imagine how the proposed regime could integrate with the existing meta-learning methods (the two extreme ends of the meta-learning in authors' conceptualization).
> > > > > I also share Reviewer onxL's concern that whether the insights could scale to larger and realistic datasets.
> > > > >
> > > > > We are not proposing that random projections are an augmentation to use for large-scale datasets. This task distribution enabled the systematic study of general-purpose meta-learning with minimal problem-specific biases. Such a systematic study is the primary purpose of the paper. For future general-purpose meta-learners we expect that high task diversity can be achieved either through natural task distributions (like it may exist in LLM-scale training data), task generation methods, or alternative problem-specific augmentation techniques. In any case, scaling general-purpose meta-learners to complex datasets is extremely difficult with current sequence models due to restrictions in context length. Given these constraints, this work is an important first step in the direction of general-purpose in-context learners.

---

### Review · Reviewer_onxL · 2023-07-18

**Summary Of Contributions:**

The paper presents an interesting contribution for the task of learning to learn based on transformers  that are general purpose learners and rely on minimal inductive biases. This is done using a transformer based architecture with carefully designed inputs that correspond to data augmentations of the input dataset. The latter are represented by very simple linear transformation of the inputs with an additional permutation of labels. The authors further provide very interesting insights into the phrase transitions of their model and the correlation of meta-test accuracy with the state size. This finding is particularly interesting as previously parameter count was considered as a main characteristic explaining the capacity of a give learner to be a general purpose learner.

**Audience:**

Yes

**Claims And Evidence:**

Yes

**Requested Changes:**

My biggest concern is regarding the scope of the insights presented in this paper. While they seem to hold for some very simple datasets, how do they carry on when considering more complex scenarios? While I do not have a strong knowledge on general purpose learning, I do have some experience in meta learning and I naturally wonder why not doing some experiments on meta learning benchmarks?

I anticipate the author's reply which may be "VSML paper considered similar setup" but in my opinion this paper will become much much stronger if some more real-world cases are considered.

My suggestions then would be to ask the authors to consider some common meta-learning benchmarks in their study, eventually in OOD settings such as miniImageNet -> CUB, or miniImagenet -> CropDisease / EuroSAT / ISIC / ChestX.

Otherwise, I really liked this work and learned a lot from reading it.

**Strengths And Weaknesses:**

**Strengths**
1. A very simple baseline for the learning of general purpose learners
2. Very insightful experimental results highlighting the different characteristics of the proposed model
3. Experimental results close to those obtained by dedicated general purpose learners with stronger inductive biases

**Weaknesses**
1. It is not clear whether the obtained results are reasonably good on the tested datasets; while the accuracy itself is of no importance for the goal pursued by the authors, it is still somewhat unclear whether the results suggests the emergence of learning to learn and just the fact that the used transformers learns low-level features well enough to be somewhat efficient on these simple datasets
2. Many insights are claimed in a very affirmative way but while reading the paper it is not entirely clear whether this amount of empirical evidence is enough for them be that strong.

---

> ### Author Response · Authors · 2023-08-14
> **Rebuttal**
>
> Thank you for taking the time to review our paper and your insightful comments. We appreciate that you really liked this work and learned a lot from reading it.
>
> ### Do the results suggest the emergence of [general] learning to learn?
> We define the process of learning simply as an increase in prediction performance with an increase in the number of examples fed into the neural network (the context). We then further qualify a general learning algorithm as one that can be applied to many different datasets. Figures 3 and 4 show that such a learning process is happening: With more examples the prediction performance increases. Table 2 and Figure 10 show that training on one base dataset generalizes to many other datasets, establishing generalization. In particular, even a randomly generated dataset that has never been seen during meta training (trained on randomized MNIST) leads to good meta-test performance. Here, such a dataset consists of 10 datapoints (x, y) where x is a random uniform vector assigned to a random label y of 10 classes.
>
> ### About insights on more complex tasks
> We agree that it would be interesting to investigate our found phenomena with more complex datasets, but doing so is currently computationally infeasible as it would require Transformers with much longer context lengths.
> We are interested in the conditions under which black-box models such as Transformers learn to implement general-purpose learning algorithms. From this it follows that if such a general-purpose learning algorithm is implemented, sample efficiency is much lower compared to ‘few-shot learners’ that are trained on a certain task distribution with the expectation that it will only learn on / generalize to distributions similar to it. Because of the general-purpose nature of the meta-learned learning algorithms, there is a limit of how good a classifier can be from just 100 examples. In Figure 14 of the appendix we showed that increasing the context length generally improves in-context learning performance. To perform well on complex datasets we would at least have to be able to attend to the entire dataset once, such as a context length of 60’000 images of CIFAR10 (or datasets like miniImageNet, CUB, CropDisease, etc), which is expensive to do with Transformers, and is currently not practically feasible. If architectural biases like the patching in Vision Transformers are desired for better test performance, this would further multiply the required context length by the numbers of patches. Increasing the context length of Transformers and related architectures is an open research question. That said, we have experimented with several different smaller datasets and the results, e.g. in terms of the algorithmic transitions in the number of tasks, seem consistent (Figure 4, 11, 12). We believe that even without more complex datasets, we have discovered interesting phenomena that are worthy of publication.
> In light of this, we are happy to tone down our claims and be more specific about the occurrence of these phenomena in our particular experimental settings.
>
> Thank you again for your review and we are looking forward to discussing any remaining concerns that you might have.

---

### Review · Reviewer_vbez · 2023-08-06

**Summary Of Contributions:**

This paper studies the use of transformers and other black-box models for meta learning:

+ Section 3 discusses their in-context learning setup. Here, a model is trained on tasks (x_i, y_i) via maximum likelihood with eq (2) where the model $f_\theta(D_{1:j-1}, x_j), y_j)$ predicts the target $y_j$ associated with $x_j$ also using knowledge from the previous tasks $D_{1:j-1}$. Alg 1 summarizes the procedure
+ Section 4 presents experimental results, mostly on MNIST/FashionMNIST/KMNIST/CIFAR10/SVHN classification. Throughout these setups, they present a number of interesting ablations and characteristics of generalization and other behaviors, such as Figs 4/5, which shows how will additional tasks help with generalization. Fig 6 shows an ablation across state and parameter sizes to show how improperly choosing them can degrade performance.

**Audience:**

Yes

**Claims And Evidence:**

Yes

**Requested Changes:**

This is a borderline paper. To meet the TMLR acceptance criteria, I request for the paper to reference and discuss how the proposed method is different from [SNAIL](https://arxiv.org/abs/1707.03141) and other papers in the citation graph around it. This is really important to contextualize the methodological contribution and connect it to existing experimental results

**Strengths And Weaknesses:**

Strengths:
1. The paper and in-context learning setup is clearly written and all of the experimental results are communicated well and easy to understand with clear takeaways.
2. The ablations and insights presented throughout the paper are interesting dimensions for practitioners to consider when meta-learning

Weaknesses:
1. [A Simple Neural Attentive Meta-Learner (SNAIL)](https://arxiv.org/abs/1707.03141) from ICLR 2017 is a significantly omitted piece of related work that also considers the use of attention-based meta-learning. Compare, e.g., figure 1 of the SNAIL paper with fig 1 of this submission.
2. Some of these settings are relatively non-standard and small-scale, e.g., Table 2 uses MNIST that trains on 99 examples and predicts the 100th. So, it is difficult to compare most of the results in this paper to other established results in the literature

---

> ### Author Response · Authors · 2023-08-14
> **Rebuttal**
>
> Thank you for taking the time to review our paper and your insightful comments. We appreciate that you like the writing of our paper, experimental investigations, and clear takeaways.
>
> ### How is this paper different from SNAIL and related work?
> SNAIL is indeed an important approach in the category of ‘in-context learning with black-box models’ (second paragraph of our related work) and we will make sure to add it to the related work. It is related to our approach in that it uses black-box neural networks and soft attention to do meta learning. The goal of our paper and the experimental setups are markedly different though: The objective of our paper is to investigate under what conditions black-box models such as Transformers learn to implement general-purpose learning algorithms.
> So in comparison to approaches like SNAIL where the authors are interested in quick adaptation to tasks that are similar to the training distribution, we are interested in much more generalizable in-context learning algorithms.
> Thus, our benchmarks also focus on much more diversity of tasks and generalization scenarios from one dataset to another, like MNIST -> Fashion MNIST / CIFAR10 / etc.
>
> ### Non-standard and small scale benchmarks
> Because of our focus on more generalizing learning algorithms, we do not use standard benchmarks that test for generalization to tasks from a more restricted task distribution (e.g. Omniglot in SNAIL).
> It is also important to note that a general-purpose learning algorithm naturally has a lower sample efficiency compared to standard ‘few-shot learners’ that are trained on a certain task distribution with the expectation that it will only learn on / generalize to distributions similar to it. To perform well on complex datasets we would at least have to be able to attend to the entire dataset once, such as a context length of 60’000 images of CIFAR10 (or datasets like miniImageNet, CUB, CropDisease, etc), which is expensive to do with Transformers, and is currently not practically feasible.
>
> Thank you again for your review and we are looking forward to discussing any remaining concerns that you might have.

---

> > ### Comment · Reviewer_vbez · 2023-08-14
> > **Response**
> >
> > Thank you, those are reasonable clarifications. With those in the paper, I've updated the responses in my review to say the paper backs up the claims and is appropriate for the TMLR audience

---

> > > ### Author Response · Authors · 2023-08-21
> > >
> > > We genuinely appreciate your valuable feedback and are thrilled by your positive response to our paper. We will add those clarifications to the paper. Thank you once again for your time and expertise.

---

### Decision · Action_Editors · 2023-09-07

**Recommendation:** Reject

**Comment:**

This paper studies a general purpose meta learning framework by using a transformer based architecture with carefully designed inputs that correspond to data augmentations of the input dataset. The setting is interesting, however, there are some issues pointed out by the reviewers that are not well addressed in the rebuttal.

Specifically, the reviewers raised a number of questions, however, except than simply replying via arguments, it seems the authors did not take a chance to carefully address them. The proposed method seems not scalable enough to,have practical values. The proposal of using random permutation of data to generate sufficiently diverse task seems to destroy valuable structural information of data. It is also unclear how generic learning to learn could be tackled with diverse and heterogeneous data. Also, the explanation for the phase shifts/emerging behaviors of the model as the task diversity increases seems not convincing enough.

I would suggest the authors to carefully revise the paper following the reviewers' comments for a potential and valuable future submission.

**Audience:**

Audience in meta learning might find the work interesting.

**Claims And Evidence:**

The claims turn out not to be well supported by enough empirical evidence, as the proposed method is not scalable enough to run on large datasets.